# Regioselective C-H Functionalization of the Six-Membered Ring of the 6,5-Fused Heterocyclic Systems: An Overview

**DOI:** 10.3390/molecules26195763

**Published:** 2021-09-23

**Authors:** Soukaina Faarasse, Nabil El Brahmi, Gérald Guillaumet, Saïd El Kazzouli

**Affiliations:** 1Euromed Research Center, Euromed Faculty of Pharmacy, Euromed University of Fes, Route de Meknes, Fez 30000, Morocco; s.faarasse@ueuromed.org (S.F.); n.elbrahmi@ueuromed.org (N.E.B.); gerald.guillaumet@univ-orleans.fr (G.G.); 2Institute of Organic and Analytical Chemistry, University of Orleans, UMR CNRS 7311, BP 6759, CEDEX 2, 45067 Orleans, France

**Keywords:** 6,5-fused heterocyclic systems, C-H functionalization, C-H/C-H activation, direct arylation, oxidative alkenylation

## Abstract

The regioselective C-H functionalization of the five-membered ring of the 6,5-fused heterocyclic systems is nowadays well documented due to its high reactivity compared to the six-membered ring. So, developing new procedures of C-H functionalization of the six-membered ring “by thinking out of the box” is extremely challenging, which explains the limited number of reports published to date. This review paper aims to highlight advances achieved in this emerging chemistry research and discusses recently reported methods.

## 1. Introduction

C-H and C-H/C-H activations, which are very attractive reactions in the field of organic synthesis, are nowadays powerful and atom economic methods for the formation of carbon–carbon bonds and thus promising (hetero)aryl−(hetero)aryl systems [1,2]. The C-H, as well as C-H/C-H activation reactions, can be divided into two sub-methods (i) direct arylation and direct alkenylation in the case of C-H activation [3] and (ii) oxidative arylation and oxidative alkenylation in the case of C-H/C-H activation [4,5]. So far, only direct arylation and oxidative alkenylation have been described for the functionalization of the six-membered ring of the 6,5-fused heterocyclic systems (Figure 1).

Although the extraordinary recent development of C-H and C-H/C-H activations of the 6,5-fused heterocycles containing at least one heteroatom on the five-membered ring, only a few examples have been reported on the regioselective functionalization of the six-membered ring. Effectively, because of the higher reactivity of the five-membered ring in contrast to the six-membered ring, its C-H activation is widely studied, while few reports have addressed the site-selective functionalization of the six-membered ring. Moreover, the site-selective C-H activation of the six-membered ring is effortlessly completed, usually after the first functionalization of one position on the five-membered ring or by the use of directing groups (DG) to control the regioselectivity [4,6]. This review paper will survey the recent development in the field with or without the use of DG [6,7,8,9,10,11,12]. Equally important, we will cover the following heterocyclic systems: indoles, indazoles, azaindoles, benzofurazans, imidazo [1,2-*a*]pyrazine, benzothiadiazole, benzoxazoles, benzothiazoles and pyrazolo[1,5-*a*]pyrimidines. In the Section 1, we will discuss the reports addressing the regioselectivity of the direct arylation on the six-membered ring of 6,5-fused heterocyclic systems without the use of DG, then with the assistance of DG. In the Section 2, we will survey the recent reports on regioselective oxidative alkenylation without the aid of DG, then with the use of DG to orient the alkenylation.

In various reported studies, the regioselectivities of C-H and C-H/C-H activations were controlled by DG such as monodentate amides [13,14], carboxylic acids [15], ketones [15,16,17], nitriles [18,19], aldehydes [15,20], phenols [21,22], esters [15,16,23], amides [15,16], *N-*methoxybenzamide [24], ureas [25,26], imines [27,28], hydrazines [29,30] and hydrazones [31], bidentate ligands [6], carbamates [32], sulfoxides [33], sulfonamides [34,35], sulfoximines [36], phosphine oxides [37,38], sulfides [39], thioamides [40] as well as some heterocycles such as purines [41], pyrimidines [42], pyrazoles [43,44], oxazolines [44], azoles [45], benzothiazoles [46], oxadiazoles [47], benzoxazoles [48,49] and 7-azaindole [50]. In some cases, heterocycles such as pyridine-*N-*oxide [51,52,53], azole-*N-*oxide [53,54], quinoline-*N-*oxide [53,55] and azaindole-*N-*oxide [56] can play both the roles of substrate and DG.

## 2. C-H Activation at C4, C5, C6 and C7 Positions of 6,5-Fused Heterocyclic Systems (Direct Arylation)

Direct arylation is a reaction of aryl halides with arenes or heteroarenes to form biaryl, aryl-heteroaryl or heteroaryl-heteroaryl products [57]. It is a very interesting and attractive alternative to traditional cross-coupling reactions such as Suzuki–Miyaura [58,59], Stille [60,61] and Negishi [62,63] because it occurs without the need for the preparation of organometallic or main*-*group reagents [2]. We will discuss in this part the site-selective arylation reactions, precisely at C4, C5, C6 and C7 positions of the six-membered rings of the 6,5-fused heterocyclic systems.

### 2.1. Direct Arylation without Directing Groups

#### 2.1.1. Benzofurazans

An elegant protocol for regioselective C4 arylation of benzofurazan 1 was developed by Doucet and collaborators [64]. A phosphine-free procedure using Pd(OAc)_2_ as a catalyst and KOAc as a base in the presence of aryl bromides as coupling partners promoted the C4 arylation on the six-membered ring of benzofurazan 1. Under the optimized conditions, the direct arylation has afforded C4-arylated products in yields ranging between 48 and 83%. Selected examples are shown in Figure 2.

When an excess of aryl bromides was used, the arylation took place at both C4 and C7 positions in a one-pot manner, leading to 4,7-diarylation of benzofurans 7–10 (Selected examples obtained in yields ranging between 57 and 72% are shown in Figure 3). Various aryl bromides with a wide range of functional groups, including fluoro, chloro, aldehyde, ketone, ester, and nitro substituents, underwent C-H arylation to afford the desired symmetrical and unsymmetrical diarylated benzofurazans in moderate to good yields.

Next, the same group has investigated the reactivity of 5-, 6- and 7-substituted benzofurazans 11, 16–17. In fact, the benzofurazan 11 substituted by electron-donating group (OMe, for example) at the C5 position underwent palladium-catalyzed direct arylation, affording two different arylation products. The arylation product at the C4 position 14 was isolated as the minor in 11% yield, while the C7 arylation product 15 was isolated as the major in 38%. In contrast, the benzofurazans substituted by electron-withdrawing groups (Br or NO_2_) either at the C6 or C7 position were not reactive under the optimal reaction conditions (Figure 4).

#### 2.1.2. Benzothiadiazoles

Marder et al. [65] have synthesized differentially substituted products by C-H arylation at the C4 and C7 positions of 2,1,3-benzothiadiazole (BT) and its monofluoro (MFBT) and difluoro (DFBT) derivatives by (hetero)aryl bromides. These synthesized products can be applied to the preparation of a variety of photonic and electronic organic materials. The developed method was effective using difluorobenzothiadiazole 18 and bromoaryls as coupling partners in the presence of Pd(OAc)_2_ (10 mol%), P*^t^*Bu_2_Me·HBF_4_ (20 mol%), pivalic acid (1 equiv.) and K_2_CO_3_ (3 equiv.) in toluene at 120 °C for 3 h. Under these reaction conditions, the expected product 19 was obtained in a 96% yield (Figure 5). With the optimized reaction conditions in hand, the authors investigated the (hetero)aryl bromide substrates’ scope of the reaction. All the tested coupling partners were smoothly transformed into the corresponding diarylated DFBT derivatives in moderate to good yields. Some selected representative examples are shown in Figure 5.

To further demonstrate the synthetic utility of this strategy, the authors used it to yield mainly monoarylated product 25 in a 76% yield. Then, 25 was functionalized by C-H activation a second time to afford differentially biarylated products 26 and 27 in 88 and 46% yields, respectively. These compounds could be relevant as donors and acceptors (D-A) structures and used as dye-sensitized solar cells (DSC) or nonlinear optical (NLO) chromophores (Figure 6a).

To extend the scope of the reaction, the Marder team has developed a procedure of a sequential double direct C-H arylation of DFBT 18, as shown in Figure 6b. Thus, after treatment with 4-bromodimethylaniline (0.8 equiv.) for 4 h under standard catalytic conditions, the monoarylated intermediate was then treated with 3-bromotoluene (1.5 equiv.), which led to the second direct arylation. Finally, the diaryl product 28 was obtained in a 65% yield (Figure 6b).

The catalytic cycle contains the following steps: (1) oxidative addition of aryl halide to the palladium (A); (2) C-H bond deprotonation (B); (3) transmetalation to generate important aryl-palladium-aryl intermediates (C); (4) reductive-elimination, which led to mono-functionalized compounds (D) (Figure 7).

#### 2.1.3. Furo[3,2-*b*]pyridines

A few examples of C7 arylations of 2,3-disubstituted furo[3,2-*b*]pyridines have been reported. In the first one published by Bertounesque et al. [66] in 2012, starting material 29 reacted with bromobenzene in the presence of Pd(OAc)_2_ (4 mol%), P(t-Bu)_2_Me·HBF_4_ (12 mol%), K_2_CO_3_ (2 equiv.) and PivOH (60 mol%) in mesitylene at 150 °C for 24 h, which led to C7 arylated compound 30 in an 88% yield. Under these optimized conditions, the scope of the C7 arylation was studied with various aryl bromides, affording the C7-arylated products in 41–92% isolated yields (only a few examples are shown in Figure 8).

It is important to note that the selective direct C7 arylation was achieved when a highly sterically hindered bromoarene was used. One example was reported using 2-bromo-1,3-dimethylbenzene, which led to C7-arylated product 34 in only 12% yield (Figure 8).

As previously reported by Fagnou and coworkers [67], the Bertounesque team has suggested that C7 arylations proceed via a concerted metallation–deprotonation mechanism (CMD) (Figure 9). They have observed that the yields of both selective C3 arylation and the subsequent C7 arylation were dramatically lower without palladium/pivalic acid co-catalyst combination.

#### 2.1.4. Pyrazolo[1,5-a]pyrimidines

A regioselective arylation reaction of pyrazolo[1,5-*a*]pyrimidines at the C7 position was achieved by Bedford’s group in 2015 [68]. This research group has shown that a phosphine-containing palladium catalyst can promote the direct arylation to the C7 position, which is the most acidic position, while a phosphine-free protocol led to C3 direct arylation. Authors have exhibited that the reaction between aryl bromide and pyrazolo[1,5-*a*]pyrimidine 35 in the presence of Pd(OAc)_2_ (10 mol%) as a catalyst, Cs_2_CO_3_ (1 equiv.) as a base, SPhos (20 mol%) as a ligand and LiCl (1 equiv.) in toluene at 150 °C led to highly selective C7-arylated pyrazolo[1,5-*a*]pyrimidine 36 (Figure 10). However, when K_2_CO_3_ was used as a base instead of Cs_2_CO_3_ and dioxane instead of toluene as a solvent in the presence of Pd(OAc)_2_ and LiCl under ligand-free conditions, the reaction led to C3-arylated products in 15 to 80% yields. Using these conditions, the scope was studied with various functional groups, including methoxy, cyano, fluoro, chloro, nitro and ester substituents, leading to different C7-arylated products in isolated yields ranging between 27 and 88% (a few examples are shown in Figure 10). It is noticed that the reaction yields were influenced by either electronic or steric factors. The reactivity at either C3 or C7 was lower when substrates containing electron-donating groups, especially fluoro or methoxy groups, were used, as well as when 2-bromo-1,3-dimethylbenzene with steric hindrance was used as an arylating agent. Authors have also shown that C3, C7-bis-arylated products can also be prepared in reasonable yields starting from either C3 or C7-arylated pyrazolo[1,5-*a*]pyrimidine (the results are not shown here).

To investigate the mechanism and regioselectivity of this reaction, the authors have carried out the deuteration experiment, which showed that the most acidic site is the C7 position. The obtained result was supported by the density functional theory (DFT) calculation (Figure 11). Additionally, the calculation of localization of the HOMO has shown that the C3 position is the most nucleophilic site. Thus, the C-H activation mechanism proposed for C3 arylation under phosphine-free conditions is likely to be an electrophilic palladation, while a base-assisted deprotonation mechanism is advanced for C7 arylation under conditions containing SPhos (Figure 11).

Wu and coworkers have reported regioselective C3 and C7 direct arylations of pyrazolo[1,5-*a*]pyrimidine 46 with aryl iodides [69]. The selectivity of the reaction is sensitive to the type of additive and solvent as well as to the presence or not of a substituent group on the C3 position. When the C3 position is occupied with ester, the C7 arylation reaction took place under the optimized reaction conditions (Pd(OAc)_2_ (5 mol%) and Ag_2_CO_3_ (2 equiv.) in a mixture of DMSO/DMF at 140 °C for 48 h).

It is worth noting that the intermediate bis(3-methylcarboxylpyrazolo[1,5-*a*]pyridine)-palladium(II) (B) was isolated, which helped the authors to propose a possible catalytic mechanism for C7 arylation. Thus, the following steps were proposed: (1) the palladium reduction, then insertion reaction into the carbon(sp2)−iodine (A); (2) the coordination reaction with 46 to give (B); (3) then metalation-deprotonation transition to give (C); (4) after that, an elimination step of silver(I) hydrogen carbonate gave (D); (5) finally, reductive-elimination reaction led to the C7-arylated product and palladium(0) (Figure 12).

#### 2.1.5. Benzoxazoles and Benzothiazoles

In 2016, Doucet et al. [70] reported a phosphine-free PdCl_2_ catalyzed C7 arylation on both benzoxazoles and benzothiazoles without the use of additional directing groups. The reactions were carried out with bromoarenes as coupling partners, PdCl_2_ (2 mol%) as catalyst and PivOK (2 equiv.) as base in NMP at 150 °C. In the case of benzoxazole 49, the arylation took place exclusively at the C7 position, leading to C7 aryl-benzoxazoles 50–55 in high yields with a broad substrate range, including sensitive functional groups such as cyano, aldehyde, ester, etc. Surprisingly, the reaction using 4-bromoanisole as an arylating agent did not lead to the corresponding arylation product. However, when 4-iodoanisole was used instead of 4-bromoanisole in the presence of 2 mol% of Xantphos as an additional ligand, the expected C7-arylated product was isolated in 53% yield. The desired products were obtained in yields ranging between 53 and 91% (only a few selected examples are shown in Figure 13).

In the case of benzothiazoles, the use of similar reaction conditions as discussed above led to the expected C7-arylated products in yields ranging between 54 and 68% (a few selected examples are shown in Figure 14).

The authors proposed a plausible mechanism of this transformation (Figure 15). It is based on oxygen chelation *N-*assisted, palladium-catalyzed, regioselective C-H bond cleavage (intermediate E), which resulted from an equilibrium between the closed (C) and the opened (D) forms of benzoxazole. A similar mechanism was proposed for benzothiazole with a sulfur chelation *N-*assisted C-H bond activation.

### 2.2. Direct Arylation with Directing Groups

#### 2.2.1. Indole

The indole ring system is one of the most studied heterocyclic systems in organic and medicinal chemistry [71,72]. Various reports have shown that the functionalization of the indoles at various positions impacts their biological properties [73]. In addition to C2 and C3 positions, which were widely studied towards C-H functionalization [74,75,76], indole has four other C-H bonds on the benzoic ring that can be functionalized. However, these C-H bonds have quasi-equivalent reactivity, which makes their functionalizations in a regioselective manner quite difficult.

##### C4 and C5 Arylations of Indoles

In 2017, during their exploration of the ortho C(sp^2^)–H functionalizations of benzaldehyde substrates using the transient directing group strategy, Yu, Zhang et al. [77] reported one example of C4 arylation of indoles in which the formyl group was installed in the C3 position of the indole 62 and the C4 arylation was achieved using Pd(OAc)_2_ (10 mol%), ligand L1 (40 mol%) and AgTFA (1 equiv.) in HFIP/TFA, which led to product 63 in 82% yield (Figure 16).

In the same year, an elegant protocol for either regioselective C4 or C5 arylation of indole was developed by Yang and collaborators using Pd and Cu as catalysts [78]. By the use of a readily available and removable pivaloyl-directing group at the C3 position of indole under the following reaction conditions (Ar-I, Pd(PPh_3_)_2_Cl_2_, Ag_2_O, DBU at 80 °C for 12 h), the C4-arylated products were regioselectively obtained in yields ranging between 58% and 83%. Then, the regioselective C5 arylation was also achieved using Ph_2_IOTf as an arylating agent, CuTc as a catalyst in the presence of dtpby in DCM at 40 °C for 12 h. In this case, the desired products were isolated in 33–68% yields. It is noticed that the arylating agent containing methoxy group at the para position has shown low reactivity, leading to the desired product 74 in a 33% isolated yield. To demonstrate the synthetic importance of their protocol, the authors have synthesized two potent inhibitors, namely, SETin-1 and tiplaxtinin, in which aryl groups are located at either the C4 or C5 position of the indole (Figure 17). The synthetic utility of these methods was demonstrated by the ready removal of the directing group for both the C4-arylated and C5-arylated products under mild reaction conditions using TsOH in glycol.

To explain this regioselectivity of the arylation reactions, the authors have proposed two catalytic cycles. In the palladium catalysis, the following steps are suggested: (1) the coordination of Pd(II) to the pivaloyl group of the substrate, followed by C-H activation at the C4 position of the indole (A); (2) the oxidative addition by ArI (B) and (3) the reductive elimination to give the C4-arylated products and regeneration of the active Pd(II) species by Ag_2_O (C). In parallel, in the copper catalysis cycle, the reactions proceed through the following steps: (1) the oxidative addition of the diaryliodonium salt to CuTc to afford the Cu (III) species (A’); (2) the coordination of the pivaloyl group (B’); (3) the aryl migration to the C5 position via a Heck-type four-membered-ring transition state (C’) and (4) the E2-type elimination to give the C5-arylated product and regenerate the active Cu catalyst (D’) (Figure 18).

Recently, Volla et al. [79] have reported a regioselective direct C4 arylation of indole and azaindole through the use of glycine as a transient directing group. Thus, using Pd(OAc)_2_ (20 mol%) and AgTFA (2 equiv.) at 110 °C in a mixture of AcOH/HFIP/H_2_O, the C4-arylated products were obtained with very good selectivity and moderate to very good yields ranging between 45 and 92%. It is noticed that substrates containing electron-donating groups have shown high reactivity compared to their corresponding iodoaryl-bearing, electron-withdrawing groups. Four selected examples are shown in Figure 19.

A plausible mechanism proposed by the authors is shown in Figure 20: (1) the condensation of (A) with glycine leads to (B); (2) the coordination of imine nitrogen and carboxylate moiety to Pd(II)-species to furnish intermediate (C); (3) the formation of intermediate (D) via a regioselective C4-H activation; (4) the oxidative addition of aryl iodide onto intermediate (D) gives (E); (5) finally, a reductive elimination provides the intermediate (F) and the action of silver trifluoroacetate affords the intermediate (G) and regenerates the Pd(II)-catalyst. The hydrolysis of (G) leads to C4-substituted indole.

##### C6 Arylation of Indoles

The first example of direct and site-selective arylation of indoles at the C6 position was reported in 2016 by Yang et al. [80]. The key to this success was the appropriate choice of the directing group (*N*–P(O)tBu_2_ (in this case) as well as the use of diaryliodoniumtriflate salts as the coupling partners in the presence of the catalytic amount of CuO.

One of the most important advantages of this method is its applicability to gram-scale synthesis. In addition, the *N-*P(O)*^t^*Bu_2_ directing group was easily removed by treatment with LiAlH_4_ to provide unprotected indole 83 in an 89% yield (Figure 21).

Based on the DFT calculations of copper-catalyzed meta-selective direct arylation of anilide using diaryliodonium salts [81], which revealed that the regioselectivity might take place following a Heck-type four-membered-ring transition state involving an aryl-Cu(III) species [82], the authors have proposed the mechanism shown in Figure 22 containing four steps. (1) the oxidative addition of diaryliodoniumtriflate to copper species (A) to afford intermediate (B); (2) the Cu species coordinates the oxygen atom of the TBPO group on indole to give complex (C); (3) the phenyl group bonded to copper is transferred to the C6 position of indole via a Heck-type four-membered-ring transition state (D), which furnishes intermediate (E) with phenyl added to the C6 position; (4) finally, a base assisted E2-type elimination gives the C6-arylated product and regenerates the active Cu catalyst.

##### C7 Arylation of Indoles

The selective functionalization at the C7 position of indoles is also very difficult to achieve and usually requires the substituents at the C2 in order to block this reactive position [83,84]. However, an efficient route including reduction of indoles to the corresponding indoline derivatives followed by C-H functionalization at the C7 position, then oxidation, successfully provided the expected C7-functionalized products (Figure 23) [85,86,87,88,89,90].

However, the development of the appropriate methods allowing the direct C-H arylation of indoles at the C7 position without additional steps of reduction and oxidation is highly needed. The use of a bulkier and more electron-withdrawing directing group on the nitrogen atom could increase the C7 selectivity of indoles (Figure 24).

After screening of various reaction conditions, the Yang group, in 2016, reported a regioselective C-H arylation at the C7 position of di-*tert*-butyl(1*H*-indol-1-yl)phosphine oxide 84 using phenylboronic acid as the coupling partner in the presence of 10 mol% of Pd(OAc)_2_ as the catalyst, 20 mol% of 2-chloropyridine L2 as the ligand and Cu(OTf)_2_, Ag_2_O and CuO as the oxidants in dioxane at 120 °C (Figure 25) [91]. In a representative example, under the above reaction conditions, the C7-phenylated product 85 was isolated in a 79% yield and with high regioselectivity (C7:C2:C3 = 96:0:4).

According to the authors, steric hindrance from the di-*tert*-butyl substituents may lead to highly limited interconversions between the oxygen–C7–H (the most favorable) and oxygen–C2–H (the least favorable) conformations. Thus, the assistance of oxygen coordination led to an excellent regioselective C7 arylation.

With the optimized conditions, the authors have investigated the substrate scope of this reaction using phenylboronic acid and various substituted indoles as coupling partners. This procedure showed a very high tolerance to various substituents on the indole ring (chlorine, ester, nitro, fluoride and methoxy groups) (Figure 26). It is also noticed that the *N-*P(O)*t*Bu_2_ directing group was easily removed by treatment with LiAlH_4_.

To study the reaction mechanism, the authors have carried out deuterated experiments. When the reaction was carried out with D_2_O in the absence of phenylboronic acid, ^1^H NMR revealed a deuteration at the C3 position, and D3 product 91 was obtained in an 87% yield. The same experiment has been carried out in the presence of 2.0 equivalents of phenylboronic acid for 12 h, which led to a 21% yield of D3 product 93 and to 72% of D3 and D7 products 92 (Figure 27). ^1^H NMR analysis of D3 and D7 product 92 revealed 68 and 26% deuterium incorporation at the C3 and C7 positions, respectively. Through these results, the authors have suggested that Pd(II) species may transmetalate with boronic acid; then, the C−H activation at the C7 position of 84 can reversibly occur with the assistance of the P(O)*t*Bu_2_ directing group.

Two years after, the same team developed another efficient rhodium-catalyzed C7 arylation of indoles using a phosphorous-directed strategy [92]. Employing aryl bromides as arylating agents containing a wide range of functional groups with various steric and electronic properties (methoxy, chlorine, mercapto, nitro, fluoride, cyano and amide groups), the arylation products were isolated in moderate to excellent yields (Figure 28). The reaction conditions were the following (Rh(PPh_3_)_3_Cl (6 mol%), LiOtBu (3 equiv.) in *m*-xylene (1 mL) at 150 °C, 24 h, under Ar). The N-P*^t^*Bu_2_ directing group of synthesized products can be easily eliminated by treatment with TsOH·H_2_O, and the deprotected compounds may be used as key intermediates in the total syntheses of indole alkaloids.

A plausible mechanism was also proposed based on two processes through the formation of a five-membered metalacycle via C–H bond cleavage at the C7 position. The first pathway I contains the following steps: (1) the formation of complex (B) by coordination of the catalytically active rhodium species (A) to the phosphorus atom of indole; (2) the generation of a rhodacycle (D) by *tert*-butoxide-assisted deprotonation at the C7 position of indole (C); (3) the oxidative addition of the aryl halide to intermediate (D) to give intermediate (E) and (4) the reductive elimination and dissociation of (F) to produce the C7-arylated product and regenerate the active catalyst (A). Meanwhile, the second pathway II involved the following steps: (1) the oxidative addition of the Wilkinson’s catalyst; (2) the base-assisted metalation of indole to give intermediate (E) and (3) the reductive elimination to furnish the C7-arylated product and regenerate the active catalyst (A) (Figure 29).

Another interesting work was reported by Larrosa and coworkers[93] for the arylation of 4-, 5-, 6-, and 7-indole carboxylic acids. The authors have shown that using two equivalents of iodoaryl, 3 mol % of [Ru(*t*BuCN)_6_](BF_4_)_2_, 2 equivalents of K_2_CO_3_, 1 equivalent of KOC(CF_3_)_3_ and 8 equivalents of *^t^*BuCN under argon (Ar) at 140 °C for 16 h, in the presence of 4-, 6- and 7-indole carboxylic acids exclusively gave the arylation at the ortho position of the carboxylic acid. The selected compounds 103–116 are shown in Figure 30.

Meanwhile, the same reaction conditions were used in the presence of H_2_O/*^t^*BuCN (1 equiv./4 equiv.) as a solvent; the 5-indole carboxylic acid 117 gave the diarylated products 118–122 in good yield (Figure 31). Interestingly, no side-arylation products at either C2-H or C3-H positions have taken place.

#### 2.2.2. Indazole

##### C7 Arylation of Indazoles

Indazoles are commonly recurring structural motifs found in numerous natural products, pharmaceuticals, and biologically active compounds, and, therefore, they are very attractive synthetic targets for medicinal chemists [94]. Driven by these very interesting biological applications, the development of mild and efficient procedures for the C-H functionalization of indazoles has attracted considerable attention from organic chemists in the last two decades.

As part of our continuous effort in developing direct arylation of heterocycles, some of us have reported the first example of direct arylation of indazole [95]. Later, we developed an original and unprecedented direct C7 arylation using 3-substituted 1*H*-indazoles containing electron-withdrawing groups on the arene ring as starting material and 4-iodotoluene as coupling partner. In the representative examples, the treatment of 123 by iodotoluene in the presence of Pd(OAc)_2_ (20 mol%), 1,10-phenanthroline (Phen) (40 mol%), K_2_CO_3_ (2 equiv.) and K_3_PO_4_ (2 equiv.) in refluxing DMA for 24 h led to the desired C7-arylated products in yields ranging between 43 and 69% (only a few examples are selected here in Figure 32). When the C3 position of 1*H*-indazole was not substituted, the arylation reaction of starting material 128 gave two different products: C3-arylated and C3/C7-diarylated products 129 and 125 in 61 and 21% yields, respectively (Figure 32) [96].

#### 2.2.3. Azaindazole

##### Direct C5 and C7 Arylation of 4-Azaindazoles

Very recently, our group [97] has developed a highly regioselective direct arylation at C5 and C7 positions of the azine ring of pyrazolo[4,3-b]pyridines (4-azaindazole). The *N-*oxide group increases the electron density of the pyridine ring, thus favoring productive π-binding interaction with the arene ring. By using the properties of the *N-*oxide group on starting material 130, we have found that by employing 5 mol% of Pd(OAc)_2_ as the catalyst, 10 mol% of 1,10-phenanthroline (Phen) as the ligand and 1.5 equivalents of K_2_CO_3_ as the base in toluene at 140 °C led to C5-arylated product 131 in a 75% yield. On the other hand, the direct arylation at the C7 position requires the use of 5 mol% of PdCl_2_ as a catalyst, 10 mol% of triphenylphosphine as ligand and 2 equivalents of K_2_CO_3_ as the base in DMA at 165 °C to generate C7-arylated product 132 in a 65% yield (Figure 33).

With the optimized reaction conditions in hand, we have explored the scope and limitations of these reactions. For the arylation at the C5 position, a variety of aryl iodides bearing different functional groups was well tolerated. The representative examples shown in Figure 34 were isolated in 54–75% yields and with complete regioselectivity for the C5 position. In addition, various aryl iodides were used as coupling partners, which led to the desired C7-arylated 4-azaindazole *N-*oxides in acceptable yields. The representative examples shown in Figure 34 were obtained in yields ranging between 60 and 67%. In both cases of C5 and C7 arylations, the electronic nature of the substrate substituents at the para position has no notable effects on the reactivity. However, the substrates with substituents at either the ortho or the meta position have shown negative consequences on the reactivity (results not shown here). In addition, the use of heteroaryl iodides as arylating agents did not lead to their corresponding arylated products at either the C5 or C7 position (results are not shown here).

#### 2.2.4. Azaindole

##### C4 Arylation of Azaindoles

In 2019, during their study of C4 direct arylation of indole using the directing group strategy, Volla et al. [79] reported some examples of C4 arylation of azaindole 141 in which the formyl group at the C3 position was assisted by glycine as an inexpensive transient directing group (Figure 35). Employing this procedure, the C4 arylations were achieved using Pd(OAc)_2_ (10 mol%) and AgTFA (2 equiv.) at 110 °C, which led to the expected products 142–144 in acceptable yields (Figure 35).

##### C6 Arylation of Azaindoles

The Pd-catalyzed direct arylation of pyridine *N-*oxides ring of 7-azaindole was reported by Fagnou et al. [56] in 2009 using aryl bromides as coupling partners (Figure 36). *N-*methyl 7-azaindole *N-*oxide 145 was regioselectively arylated at the C6 position using Pd(OAc)_2_ (4 mol%), DavePhos (15 mol%), PivOH (30 mol%) and Cs_2_CO_3_ (2 equiv.) in toluene at 110 °C. A wide variety of functional groups such as methyl, methoxy, bromo, fluoro or trifluoromethyl, and even ester on the aryl bromide were tolerated. The representative examples of the C6-arylated products shown in Figure 36 were obtained in yields ranging between 46 and 87%. It is noticed that the reactivity was lower because of the steric hindrance when 2-bromotoluene was used as an arylating agent.

##### C7 Arylation of Azaindoles

In the same report [56] and under similar reaction conditions, the direct C7 arylation of *N-*methyl 6-azaindole *N-*oxide 150 led to the expected products in good yields (Figure 37). Additionally, a range of aryl bromides has been shown to undergo regioselective coupling at C7 of 150, leading to the desired products 151–154 in yields ranging between 55 and 70%.

#### 2.2.5. Imidazo[1,2-*a*]pyrazine

##### C5 Arylation of imidazo[1,2-*a*]pyrazines

Snieckus and coworkers [98] have succeeded in achieving a palladium-catalyzed, regioselective C5 arylation of imidazo[1,5-*a*]pyrazine. This general method allowed the synthesis of diverse substituted derivatives in good to excellent yields by the coupling of a variety of imidazo[1,5-*a*]pyrazines 155–158 bearing either methoxy, amino, dimethylamino (NMe_2_) or aryl group at the C8 position with aryl bromides as an arylating agent in the presence of Pd(OAc)_2_/PtBu_2_CH_3_.HBF_4_ (0.1 equiv./0.2 equiv.) and Cs_2_CO_3_ (3 equiv.), in DMF at 120–130 °C. The desired products were obtained in good HPLC yields using 4-methoxy-benzoic acid as an internal standard. With these optimized conditions in hand, the authors have carried out various C5 arylations of imidazo[1,5-*a*]pyrazines with aryl halides. The expected products were obtained in good yields ranging between 45 and 95% (Figure 38). Cyano- and nitro-substituted aryl bromides are not applicable due to their decomposition.

The catalytic cycle shown in Figure 38 contains the following steps: (1) the rate-determining carbopalladation; (2) the formation of an π-aza-allyl intermediate; and (3) the reductive elimination to give the desired C5-arylated compound.

##### C6 Arylation of imidazo[1,2-a]pyrazines

In 2012, Guchhait et al. [99] reported a procedure for the C6 arylation of versatile substituted 3-aminoimidazo[1,2-*a*]pyrazines. They observed that the reaction between 164 and bromotoluene in the presence of K_2_CO_3_ (2 equiv.), PPh_3_ (20 mol%) and PivOH (30 mol%) with Pd(OAc)_2_ (10 mol%) as the catalytic system in toluene at 110 °C were the best reaction conditions. The desired C6-arylated compound 165 was isolated in a 65% yield. It is noticed that this procedure tolerated a variety of functional groups on the aryl bromide, such as ester, formyl, acetyl, nitrile, nitro and fluoro. The desired products 166–169 (only a few examples are shown here) were obtained in yields ranging between 46 and 65% (Figure 39).

A CMD mechanism proposed by the authors is shown in Figure 40 and can be divided into the following steps: (1) the oxidative addition of aryl bromide to Pd(0), which led to ArPdBr (A); (2) the reaction with PivOK to generate the ArPd(OPiv) complex (B); (3) this complex induced C6–H bond activation, which gave complex (C); (4) the production of aryl-Pd-imidazopyrazine intermediate (D) and (5) the reductive elimination to furnish C6-arylated imidazopyrazine and regenerate Pd(0) (E).

In 2014, Huestis and Johnson [100] developed a new procedure of double direct C-H arylation of imidazo[1,2-*a*]pyrazine 170 at the C3 position and then at the C5 position. Thus, after treatment with 5-bromopyrimidine under the following reaction conditions (Pd(OAc)_2_, PCy_3_.HBF_4_, PivOH, K_2_CO_3_, 100 °C), the desired monoarylated product 171 was isolated in a 60% yield. Then, C5 arylation reaction was carried out on 171 using *p*-bromoethylbenzene in the presence of Pd(OAc)_2_ (10 mol%), 1,10-phenanthroline (20 mol%), Cs_2_CO_3_ (3 equiv.) in DMA at 140 °C under the atmospheric air which furnished the bis-arylated product 172 in a 50% yield (Figure 41).

This procedure has shown to be tolerant towards many useful functional groups on the aryl bromides, such as methoxy, fluoro or trifluoromethyl functions. In addition, the developed method also allowed the introduction of various heteroaryls at both C3 and C5 positions. In these cases, the desired products 172–176 were isolated in low to acceptable yields (24–50%) (Figure 42).

#### 2.2.6. Benzoxazole

##### C7 Arylation of Benzoxazoles

It was well established that when using benzoxazolyl as a directing group, the direct arylations of 2-arylbenzoxazoles took place at the ortho position of the aryl ring [45,46,48]. To address this limitation and provide access to C7-arylated benzoxazoles, the presence of an additional directing group is required.

Cho et al. [101] have reported a palladium-catalyzed direct arylation of 2-arylbenzoxazole 177 at C7 position in a 45% yield using Ph-Br, Pd_2_(dba)_3_, X-Phos and K_3_PO_4_ in toluene at 100 °C. This uncommon regioselectivity probably accrued because of the presence of the hydroxy group at the C6 position, which acts as an ortho-directing group (Figure 43).

## 3. Oxidative Alkenylation at C4, C5, C6 and C7 Position of 6,5-Fused Heterocyclic Systems (C-H/C-H Activation)

### 3.1. Oxidative Alkenylation without Directing Groups

#### 3.1.1. Indazole

##### C7-Oxidative Alkenylation

In 2015, some of us [102] reported the first example of regioselective C7-oxidative alkenylation of the six-membered rings in 6,5-heterocyclic systems 178 without directing groups. In fact, the use of Pd(OAc)_2_ (5 mol%) as the catalyst and Ag_2_CO_3_ (2.5 equiv.) as the oxidant promoted the C7 alkenylation of substituted (1*H*)-indazoles, affording the desired products 179–183 in yields ranging between 49 and 74% (only a few selected examples are shown in Figure 44). It is noticed that the C7-oxidative alkenylation did not take place regioselectively at C7 position when R’ = H. Interestingly, when 1-methyl-5-nitroindazole 184 was treated by cyclohexyl acrylate, the reaction led to the C7-alkenylated product 185 in a 65% yield, along with the C3, C7-dialkenylated product 186 in a 17% yield (Figure 44).

### 3.2. Oxidative Alkenylation with Directing Groups

#### 3.2.1. Indole

##### C4-Oxidative Alkenylation

The regioselective C-H functionalization at the C4 position of indoles is extremely difficult because of the lack of the nucleophilic reactivity of the azine ring along with the high nucleophilic reactivity of the pyrrole moiety. However, the use of directing groups was crucial in achieving this functionalization by the development of successful approaches.

Jia et al. [103] have developed the first method, catalyzed by Pd, for the direct olefination at the C4 position of tryptophan derivatives via C-H/C-H activation using TfNH- as a directing group. The reactivity and the selectivity at the C4 position of tryptophan were improved using the protection of indole nitrogen with a bulky group such as TIPS. The regioselective olefination of tryptophan 187 at the C4 position was achieved under the following reaction conditions (Pd(OAc)_2_, AgOAc in toluene at 100 °C), which furnished the 4-substituted tryptophan derivative 188 in a 88% yield.

With the optimized reaction conditions in hand, the authors have investigated the substrate scope and the limitation. The tryptophan was treated by various olefins to afford the expected C4-olefinated products in 29–88% yields (some examples are presented in Figure 45). The presence of the methyl group at the C2 position of the olefin had a negative effect on the reactivity, and the desired product 191 was obtained in a lower yield. It is noticed that the presence of functional groups on the olefins such as ester, ketone, amide, nitrile, benzenesulfonyl and dimethoxy phosphoryl groups did not affect the reaction yields (the results are not shown).

To highlight the direct functionalization methodology at the C4 position of tryptophan derivatives, the authors have carried out the biomimetic synthesis of an alkaloid, clavicipitic acid, induced by indole tryptophan hemiterpene (Figure 46).

Under mild reaction conditions, Prabhu et al. [104] have developed a selective functionalization of indole at the C4 position by employing the aldehyde functional group at the C3 position as a directing group. The authors have found that the reaction involved a six-membered transition state to lead to the desired products. Thus, by using [Ru(p-cymene)Cl_2_]_2_ (10 mol%), Ag salt (20 mol %), Cu(OAc)_2_.H_2_O (0.5 equiv.) and 196 as starting material in ClCH_2_CH_2_Cl at 120 °C in an open flask under atmospheric air, the expected C4-alkenylated indole 197 was isolated in 70% yield (82% NMR conversion) (Figure 47).

With the optimized reaction conditions in hand, the authors have explored the scope and limitations of this protocol. Thus, a variety of 1-benzyl-1*H*-indole-3-carbaldehyde derivatives were treated with methyl acrylate to furnish C4-alkenylated products 198–200 in moderate to good yields (35 to 92%). The presence of a chloro substituent at the C6 position has shown a negative effect on the substrate reactivity (Figure 47).

Four years later, Yanxing Jia et al. [105] reported a new method for direct C4 functionalization of unprotected indoles. The best reaction conditions were found to be [RhCp*Cl_2_]_2_ (2.5 mol %), AgSbF_6_ (10 mol %) and Cu(OAc)_2_ (0.6 mmol) in DCE at 110 °C for 4 h. Under these conditions, the desired C4-alkenylated product 202 was isolated in a 51% yield. The authors have shown the versatility of this direct C-H/C-H activation on 1-*H*-indole-3-carboxaldehyde 201 by the synthesis of various derivatives in moderate to very good yields using a variety of acrylates and indole-3-carboxaldehyde derivatives as reaction partners. In contrast to the acrylates, the styrene used as coupling partner showed low reactivity (compound 207, 35% yield) (only four representative examples are shown in Figure 48).

The potential of the present C-H activation strategy was demonstrated by the first catalytic asymmetric total synthesis of (−)-agroclavine (1) and (−)-elymoclavine.

Prabhu and coworkers [106] have reported the role of the directing group at C3 position in the direct alkenylation of 1-methylindole derivatives 209. The choice between the five-membered and the six-membered ring for the alkenylation is influenced by the nature of the directing group. After a thorough investigation, the good directing group and the best catalytic system to achieve the alkenylated products 210–222 were found to be COCF_3_ as the directing group, [RhCp*Cl_2_] (2.5 mol%) as the catalyst and AgSbF_6_ (10 mol%) and Cu(OAc)_2_. H_2_O (1 equiv.) as additives in DCE at 100 °C (a few examples are shown in Figure 49).

Zhang and coworkers [107] have described the ability to perform a dual one-pot C–H bond alkenylation of indole-3-carboxylic acids 223 with alkenes catalyzed by rhodium. The subsequent decarboxylation of the directing group provides direct access to 2,4-dialkenylindoles 224–230 in good overall yields. The optimized reaction conditions were [Cp*RhCl_2_]_2_ (5 mol%), AgSbF_6_ (2.1 equiv.) and KOAc (1 equiv.) in CH_2_Cl_2_ at 100 °C (Figure 50). Under these conditions, a variety of acrylates and *N*-alkyl indoles were tolerated, except the free (NH) indoles, which are inactive under these conditions. This method has provided fluorescent alkenylindoles, which have a good fluorescence performance in the range of 472–530 nm, with a quantum yield up to 0.57 in CH_2_Cl_2._

In 2019, Ravikumar and coworkers [108] reported the direct C4-alkenylation of 3-acetylindole 231, with a variety of Michael acceptors that gave C4-alkenylated indoles 232 (Figure 51). After a thorough investigation of the influence of the solvent, the base, the nature of acrylates and the nature of the directing groups on the selectivity of the reaction, the best catalytic system was found to be the following (Cp*Co(CO)I_2_ (10 mol%), AgSbF_6_ (20 mol%), with Cu(OAc)_2_.H_2_O (1 equiv.) in HFIP (0.1 M) at 50 °C for 20 h).

Subsequently, the scope of the reaction was studied utilizing *N*-substituted maleimides as a coupling partner under the optimized reaction conditions. In this case, it was demonstrated that by switching the Cu(OAc)_2_.H_2_O to Ag_2_CO_3_, the alkylenation also occurs well at the C4 position to give the indoles 233–238. In the case of the substrate bearing a methoxy group at the C5 position, only traces of C4-alkenylated product 237 were observed (Figure 52; only representative examples are shown).

A catalytic cycle has been proposed involving the reaction of Cp*Co(CO)I_2_ with AgSbF_6_ to generate an active catalyst I, followed by cyclometalation with (A) to form a six-membered species II. The cationic cobalt(III) species with Michael acceptor (B) gives the intermediate III by π-complexation followed by olefin insertion, which produced the intermediate IV. Finally, β-hydride elimination followed by reductive elimination afforded the final product (C) and cobalt(I) complex (V), which is oxidized by Cu(OAc)_2_ to generate the catalyst cobalt(III) I for the next catalytic cycle (Figure 53).

##### C5-Oxidative Alkenylation

In 2011, Gloriusand et al. [109] reported one example of C5-oxidative alkenylation of indole 239 catalyzed by rhodium. This reaction was oriented by the ketone (COMe) at the C6 position, which plays the role of the directing group. The reaction was carried out using [(Cp*RhCl_2_)_2_] (0.5 mol%) in the presence of AgSbF_6_ (2 mol%) and Cu(OAc)_2_ (2.1 equiv.) in t-AmylOH at 120 °C for 16 h. Under these conditions, the desired C5-alkenylated indole 240 was obtained with very high selectivity (Figure 54).

##### C6-Oxidative Alkenylation

The same group also described one example of C6 oxidative alkenylation of indole 241 catalyzed by rhodium under the same reaction conditions discussed above [110]. In this case, the directing group (COMe) was at the C5 position, leading to regioselective C6 alkenylation 242 in very good selectivity (Figure 55).

In 2014, Yu et al. [84] reported C-H olefination of indoles at C6 position using removable sulfonamide linked to N1 position. The authors developed the following reaction conditions (ethylacylate in the presence of 10 mol% of Pd(OAc)_2_, 20 mol% of Ac-Gly-OH, three equivalents of AgOAc in hexafluoroisopropanol (HFIP) at 70 °C for 24 h. In the representative examples, starting materials 243, 244 and 245 were treated by ethyl acrylate under the conditions cited above to lead to C6-alkenylated products 246, 247 and 248 in 66, 58 and 49% yields, respectively (Figure 56).

##### C7-Oxidative Alkenylation

In 2013, Song’s group [111] reported an efficient atom-economic one-pot approach for the preparation of C7-substituted indoles via rhodium(III)-catalyzed oxidative cross-coupling. Regioselective olefination of indoline derivative 249 using (Cp*RhCl_2_)_2_ (5 mol%), AgSbF_6_ (20 mol%), Cu(OAc)_2_ (2.5 equiv.) and *t*-AmOH at 120 °C followed by subsequent one-pot oxidation using MnO_2_ (30 equiv.) at 120 °C provided the desired product 250 in an 80% yield (Figure 57).

Having the optimized reaction conditions in hand, the authors explored the scope and limitation of this reaction. A series of alkene derivatives with various substituents were subjected to olefination with indole, which afforded the desired products in moderate to good yields (Figure 57).

The mechanistic study proposed by the authors suggested that the C7-alkenylation of indoles was promoted by a catalytic dehydrogenative cross-coupling of indoline followed by its subsequent oxidation (Figure 58).

In order to demonstrate the additional synthetic utility of their approach, the authors have achieved C2 alkenylation of 250 with different olefins (in the representative example, the compound 255 was isolated in a 63% yield). Additionally, the hydrolysis of 250 by NaOH (aq) afforded C7-substituted NH-free indole 256 in an isolated yield of 88%. Finally, the hydrogenation on Pd/C of 250 led to C7-alkylated indole 257 in a 72% yield (Figure 59).

By exploring the functionalization of the C7 position of indole, the Wei Yi group [112] have reported the first instance of a Rh(III)/Cu(II)-catalyzed direct C7 alkenylation of indole C2-alkylated. In fact, a series of indole derivatives 258 with various substituents were subjected to coupling with α-diazotized Meldrum’s acid in EtOH to give the corresponding C2 ethyl-acetate-substituted indoles 259 in good to excellent yields (Figure 60). Then, the compounds 259 were coupled with methyl acrylate by employing the transition metal-catalyzed C-H activation strategy under the following conditions ([Cp*RhCl_2_](5 mol%), Cu(OAc)_2_, H_2_O (1 equiv.), DMF, 80 °C, 6 h). In the representative examples, one unsubstituted indole and two other substituted indoles at the C5 position with either methyl or methoxy group were treated under the optimized reaction conditions offering the desired products 261–263 in good yields (Figure 60).

During their development of direct arylation of indoles at the C7 position, Yang and his group [91] have explored a directing group strategy to achieve oxidative alkenylation at the C7 position. They used methyl acrylate and indoles 264 as coupling partners in the presence of Pd(OPiv)_2_ (10 mol%) as the catalyst, L3 (20 mol%) as the ligand and Cu(OTf)_2_ (0.55 equiv.) and CuO (1 equiv.) as oxidants under O_2_ in DCE at 80 °C for 24 h, which afforded the desired C7 olefination products 265–267 in good to excellent yields. The presence of a methyl group at either C3 or C4 position led to good reaction yields as well as high regioselectivities (Figure 61).

During their studies on the functionalization of the non*-*naturally amino acids by C-H activation [110,113], Ma and his team discovered that the regioselective C7 olefination of indoles could be promoted by using rhodium as catalyst and *N-*pivaloyl as a directing group [114]. Thus, in 2016, they reported an effective method for C7 olefination of *N-*pivaloylindole 268 with methyl acrylate, in the presence of (Cp*RhCl_2_)_2_ (4 mol%) as a catalyst and AgNTf_2_ (16 mol%) and Cu(OAc)_2_·H_2_O (2.1 equiv.) in CH_2_Cl_2_, at 80 °C for 36 h. Under these reaction conditions, the expected product 269 was obtained in an 83% yield (Figure 62).

The substrate scope of the reaction was explored with various indoles containing various substituents at either the C3 or C5 position. Methyl acrylate was used as olefin to furnish the corresponding C7-olefinated products in good to excellent yields (68–95%). Electron*-*rich indoles were very reactive under these reaction conditions. In contrast, under similar conditions, the substrate with the Cl group at the C6 position of indole was not efficient. It is noticed that aliphatic acrylate derivatives, namely, methyl, butyl and *n-*butyl, and aromatic derivatives, such as phenyl, were found to furnish good to excellent yields (results are not shown here).

The treatment of 269 with triethylamine in methanol led to the cleavage of the Boc-protecting group while the methyl ester remained intact, leading to easy removal of the directing group and providing the C7-functionalized tryptophan derivative 274 in a 94% yield (Figure 62).

#### 3.2.2. Indazole

##### C7-Oxidative Alkenylation

Very recently, Pan et al. [115] have developed the first example of Rh(III)-catalyzed regioselective C7 functionalization of 1*H*-indazole using directing groups. After examining several coordinating directing groups in the reaction of 1*H*-indazole with methyl acrylate, pivaloyl, acyl, 2-pyrimidy, Boc and methanesulfonyl groups were not effective. The use of CONMe_2_ and CONMeEt as directing groups led to their corresponding C7 products in 24 and 32% yields, respectively, while the use of CONnHex_2_ provided a 70% yield of the desired product. The best result was observed when using *N*,*N-*diisopropyl carbamoyl indazole. The product was obtained in an 85% yield showing that increasing the size of the substituted urea group is very important (Table 1, entries 1–5).

With the coupling of 275 and methyl acrylate as a model reaction, a range of catalysts, oxidants and solvents were further tested to improve the reaction yield. The optimized reaction conditions were found to be the following ([Cp*RhCl_2_]_2_ (Cp* = 1,2,3,4,5-pentamethylcyclopentadienyl) (4 mol%), AgSbF_6_ (16 mol%) and AgOAc (0.42 mmol) in BuOH at 80 °C for 48 h). Under these conditions, the desired C7 olefinated indazole 276 was isolated in a 90% yield. With the optimized reaction conditions in hand, a series of substituted 1*H*-indazoles were then employed to investigate the scope and limitation as well as the functional group tolerance of this protocol. The presence of the bromo group at the C6 position of the indazole had a negative effect on the substrate reactivity, and thus the expected alkenylated product was isolated in a very low yield (compound 279). In fact, various C7-alkenylated products were prepared in low to excellent yields (18–94% yields). Some selected representative examples, 277–280, are shown in Figure 63.

## 4. Conclusions

In this review, we have presented an overview of the C-H functionalization of the six-membered rings of the 6,5 fused heterocyclic systems containing heteroatoms (N, O, S) via the C-H bond activation process. Numerous new methods towards C-H and C-H/C-H activations have been developed and have greatly impacted the synthetic strategies, and a number of competing approaches for the control of the regioselectivity of those reactions have emerged. Thus, the elegant methods have been recently developed with or without the assistance of the directing groups; despite this, there are still some major challenges that need to be addressed in this field. For example, in the most reported reaction conditions, noble metals such as palladium, ruthenium and rhodium are essential to serve as catalysts. Furthermore, some C-H or C-H/C-H activation reactions still suffer from some drawbacks, such as poor regioselectivity as well as limitation of substrates’ scope. It should be noted that the limitation of regioselectivity often required the use of DG to achieve C-H or C-H/C-H functionalization on the six-membered rings. However, the use of DG usually led to the functionalization at only the α- or β-positions and sometimes needed to be removed, which recurred additional treatments (additional reactions and work-up). Currently, the development of innovative reaction conditions to enhance regioselectivity is still needed. Undoubtedly, the recent developments will help both synthetic and material chemists to make C-H and C-H/C-H functionalization a valuable tool for diverse academic and industrial applications.

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
