# Peer review of "Regioselective C-H Functionalization of the Six-Membered Ring of the 6,5-Fused Heterocyclic Systems: An Overview"

_molecules, 2021, doi:10.3390/molecules26195763_

Round 1

Reviewer 1 Report

Saïd El Kazzouli and coworkers presented a review article on the regioselective C-H functionalization of the six membered ring of the 6,5-fused heterocyclic systems, as a review, it is very helpful for researchers or graduate students to master various reported methods for the functionalization of benzenoid ring of 6,5-fused heterocycles. However, the review is not well prepared before submission. Although the authors did a good job in literature search and tried to incorporate as many reactions as possible, there is no particular rationale or order in which the heterocyclic systems were reported. The introduction provided is not sufficient enough to explain the necessity of this review. The authors also did not provide any insight about the reactivity of these heterocycles and why the reactivity of 5-membered ring is different from that of the 6-membered ring. Other than saying these were all the reactions reported on these heterocycles and here is the mechanism reported by original authors, the review doesn’t provide authors insights and seems like written by younger researchers. At the current stage, this review needs a major rewrite before considering for the publication.

Few minor comments:

  1. Be consistent with reporting reaction conditions in the discussion – either report loading of the reagents for all the reactions or don’t report loadings

Ex: page 3, line 95 “in the presence of Pd(OAc)2, PtBu2Me·HBF4, pivalic acid and K2CO3 in toluene at 120 °C for 3 h” vs page 5, line 127 “Pd(OAc)2 (4 mol%), P(t-Bu)2Me·HBF4 (12 mol%), K2CO3 (2 equiv) and PivOH (60 mol%) in mesitylene at 150 °C for 24 h

  1. Page 9, Scheme 16 and line 227 why is the ligand numbered L4, I don’t see any L1-L3 in the manuscript prior to this – authors might have just used the ligand number published in the original manuscript – please correct this use, and use numbering appropriately throughout the review
  2. Same issue in Scheme 25 – please check the manuscript thoroughly
  3. Page 9, line 227 – AgFTA should be changed to AgTFA
  4. Page 6, line 159-161 – please explain or expand the statement “Authors have also confirmed that the C3 or C7-arylation reaction products can then be arylated in the alternative position with reasonable yields.”
  5. Page 6, Scheme 9 – no numbers to the compounds, and in the scheme title
  6. Please check for spelling mistakes throughout the manuscript – there are lot of them
  7. Page 1, Scheme 1 – it is confusing, please redraw to accurately describe the review article
  8. For all the schemes and discussions, please add a space between number and °C (120°C vs 120 °C)
  9. Page 23, line 529 – compound 185 is not C3-alkenylated compound
  10. Discussion for every reaction doesn’t need to have optimized conditions listed rather discuss the impact of the original manuscript on the overall development of the field
  11. For the schemes, when the representative substrate scope is shown, please highlight the reactivity of the substrate electronics or sterics and/or any factors influencing the mentioned reactivity rather than just showing the representative examples with no particular emphasis on any other information
  12. Page 33, line 747 – I don’t understand what does authors mean by “main mechanisms of organic chemistry”
  13. In general, I don’t see the point of section 5 – “overall picture”, the generalized mechanisms in Scheme 64 are wrong depictions and they should be either completely deleted or modified accordingly

Author Response

Reviewer 1.

Saïd El Kazzouli and coworkers presented a review article on the regioselective C-H functionalization of the six membered ring of the 6,5-fused heterocyclic systems, as a review, it is very helpful for researchers or graduate students to master various reported methods for the functionalization of benzenoid ring of 6,5-fused heterocycles. However, the review is not well prepared before submission. Although the authors did a good job in literature search and tried to incorporate as many reactions as possible, there is no particular rationale or order in which the heterocyclic systems were reported. The introduction provided is not sufficient enough to explain the necessity of this review. The authors also did not provide any insight about the reactivity of these heterocycles and why the reactivity of 5-membered ring is different from that of the 6-membered ring. Other than saying these were all the reactions reported on these heterocycles and here is the mechanism reported by original authors, the review doesn’t provide authors insights and seems like written by younger researchers. At the current stage, this review needs a major rewrite before considering for the publication.

Thank you reviewer 1 for this comment, the introduction was revised (see revised version)

Few minor comments:

  1. Be consistent with reporting reaction conditions in the discussion – either report loading of the reagents for all the reactions or don’t report loadings

Ex: page 3, line 95 “in the presence of Pd(OAc)2, PtBu2Me·HBF4, pivalic acid and K2CO3 in toluene at 120 °C for 3 h” vs page 5, line 127 “Pd(OAc)2 (4 mol%), P(t-Bu)2Me·HBF4 (12 mol%), K2CO3 (2 equiv) and PivOH (60 mol%) in mesitylene at 150 °C for 24 h

Thank you for this comment, this was done (see revised version)

  1. Page 9, Scheme 16 and line 227 why is the ligand numbered L4, I don’t see any L1-L3 in the manuscript prior to this – authors might have just used the ligand number published in the original manuscript – please correct this use, and use numbering appropriately throughout the review

Thank you for this comment, all the ligands cited in this review were numbered L1, L2 … in a chronologic manner (see revised version).

  1. Same issue in Scheme 25 – please check the manuscript thoroughly

Thank you for this comment, all the ligands cited in this review were numbered L1, L2 … in a chronologic manner (see revised version).

  1. Page 9, line 227 – AgFTA should be changed to AgTFA 

This was done (see revised version).

  1. Page 6, line 159-161 – please explain or expand the statement “Authors have also confirmed that the C3 or C7-arylation reaction products can then be arylated in the alternative position with reasonable yields.”

More details are added to explain this statement (see the revised version).

  1. Page 6, Scheme 9 – no numbers to the compounds, and in the scheme title:

This was done (see revised version).

  1. Please check for spelling mistakes throughout the manuscript – there are lot of them

Thank you for this comment, our manuscript was checked again by an English native speaker.

  1. Page 1, Scheme 1 – it is confusing, please redraw to accurately describe the review article

Thank you for this comment, this scheme was modified (see the revised version)

  1. For all the schemes and discussions, please add a space between number and °C (120°C vs 120 °C)

This was done (see revised version)

  1. Page 23, line 529 – compound 185 is not C3-alkenylated compound

This was done (see revised version)

  1. Discussion for every reaction doesn’t need to have optimized conditions listed rather discuss the impact of the original manuscript on the overall development of the field

Most of the discussed reports are very important and often they are pioneers. For this reason, we thought that it is important to discuss the optimized reaction conditions. When the original manuscript has very important impact on the development of the field we gave more explanations and more details on the reaction scope and limitations. See the revised version.

  1. For the schemes, when the representative substrate scope is shown, please highlight the reactivity of the substrate electronics or sterics and/or any factors influencing the mentioned reactivity rather than just showing the representative examples with no particular emphasis on any other information

In some cases, we did not mention that, however, when the reactivity is clearly depending on the substrate electronics or sterics and/or any factors we mentioned that. In the revised version we addressed most of these points.

  1. Page 33, line 747 – I don’t understand what does authors mean by “main mechanisms of organic chemistry”

This section was deleted

  1. In general, I don’t see the point of section 5 – “overall picture”, the generalized mechanisms in Scheme 64 are wrong depictions and they should be either completely deleted or modified accordingly

This section was deleted

Reviewer 2 Report

This manuscript summaries recent achievements of the C–H functionalization of the six-membered rings of the 6,5-fused heterocyclic systems via process of C–H bond activation. Numerous heterocycles such as benzofurazanes, benzothiadiazoles, pyrazolo[1,5-a]pyrimidines and indoles are shown in modification by this strategy, allowing for incorporating aryl or alkenyl into the scaffolds in accessing medicinally interesting derivatives. This review is timely and almost covers recent developments in this arena. The publication of this Review in Molecules is strongly commended after addressing the following problems.

(1) Scheme 1, please replacement of “X” in the left molecule scaffold by “Z”, because it clearly demonstrates the term of“Z = N, O or S”.

(2) Page 1, please using the form of italics for“a”in“[1,2-a]”. Similar mistake in page 1, line 37 should be corrected.

(3) Page 2, please replacement of “pyridines-N-oxide”by “pyridine-N-oxide”.

(4) Scheme 2, the title for the Scheme, please using the bold form for “1”. Similar problem in Scheme 3 should be corrected.

(5) Page 3, line 80, please revising the“electron n-donating”by “electron-donating”, and it’s better to replacement of “OCH3”by “OMe”in the sentence.

(6) Page 3, line 83, please replacement of “manor”by “minor”.

(7) Page 3, line 85, it’s better to replacement of “the developed reaction conditions”by “the optimal reaction conditions”.

(8) Page 3, line 91, please using the form of italics for “et al.”.

(9) Page 5, line 125, please using the form of italics for “b”in “[3,2-b]”. Similar problems are found in lines 126, 147, 153, 154, 165… for“[1,5-a]”.

(10) Scheme 11, please showing the unit for “calculated energies”.

(11) Page 7, line 177, please replacement of “idodides” by “iodides”.

(12) Page 7, line 180, please revising “AgCO3”by “Ag2CO3”.

(13) Line 181, please showing the blanks for “140oC” and “48h”.

(14) Line 201, please using the form of bold for compound of“49”.

(15) Scheme 16 and line 228, please correction of “AgFTA”by “AgOTf”. Similar error was found in line 260.

(16) Scheme 17 and line 237, whether should the ligand of “dtpby”be “dtbpy”?

(17) Line 264, please using the bold form for compound of “76”. Similar error was found in line 295 for compound of “81”.

(18) Line 326, please replacement of “fluor”by “fluoride”. Similar error was found in line 345.

(19) Line 348, please revising “ml”by “mL”.

(20) Line 368, please replacement of “idoaryl”by “iodoaryl”.

(21) Line 391, please replacement of “an electron withdrawing groups”by “an electron-withdrawing group”.

(22) Line 431 and Scheme 35, “AgTFA”should be “AgOTf”.

(23) Line 436, please removal of the blank between “direct”and “arylation”.

(24) Line 513, removal of the blank between “using”and “Ph–Br”, and replacement of “Pd2dba3”by “Pd2(dba)3”.

(25) Line 566, please revising the “Cu(OAc)2.3H2O”. Similar error was found in Line 593, 612 and 618, and Schemes 49 and 50.

(26) Lines 681–691, please showing the compounds 258, 259, 261–263 by the bold form.

(27) Page 33, Table 1, please revising “Yield%”by “Yield (%)”.

(28) Line 787, it seems that some terms are missing. Please revising this sentence.   

Author Response

Reviewer 2.

This manuscript summaries recent achievements of the C–H functionalization of the six-membered rings of the 6,5-fused heterocyclic systems via process of C–H bond activation. Numerous heterocycles such as benzofurazanes, benzothiadiazoles, pyrazolo[1,5-a]pyrimidines and indoles are shown in modification by this strategy, allowing for incorporating aryl or alkenyl into the scaffolds in accessing medicinally interesting derivatives. This review is timely and almost covers recent developments in this arena. The publication of this Review in Molecules is strongly commended after addressing the following problems.

  • Scheme 1, please replacement of “X” in the left molecule scaffold by “Z”, because it clearly demonstrates the term of“Z = N, O or S”.

Thank you reviewer 2 for this comment, this scheme was modified (see the revised version)

  • Page 1, please using the form of italics for“a”in“[1,2-a]”. Similar mistake in page 1, line 37 should be corrected.

This was done (see revised version)

  • Page 2, please replacement of “pyridines-N-oxide”by “pyridine-N-oxide”.

 This was done (see revised version)

  • Scheme 2, the title for the Scheme, please using the bold form for “1”. Similar problem in Scheme 3 should be corrected.

This was done (see revised version)

(5) Page 3, line 80, please revising the“electron n-donating”by “electron-donating”, and it’s better to replacement of “OCH3”by “OMe”in the sentence.

This was done (see revised version)

  • Page 3, line 83, please replacement of “manor”by “minor”.

This was done (see revised version)

  • Page 3, line 85, it’s better to replacement of “the developed reaction conditions”by “the optimal reaction conditions”.

This was done (see revised version)

  • Page 3, line 91, please using the form of italics for “et al.”.

This was done (see revised version)

  • Page 5, line 125, please using the form of italics for “b”in “[3,2-b]”. Similar problems are found in lines 126, 147, 153, 154, 165… for“[1,5-a]”.

This was done (see revised version)

  • Scheme 11, please showing the unit for “calculated energies”.

This was done (see revised version)

  • Page 7, line 177, please replacement of “idodides” by “iodides”.

This was done (see revised version)

  • Page 7, line 180, please revising “AgCO3”by “Ag2CO3”.

This was done (see revised version)

  • Line 181, please showing the blanks for “140oC” and “48h”.

This was done (see revised version)

  • Line 201, please using the form of bold for compound of“49”.

This was done (see revised version)

  • Scheme 16 and line 228, please correction of “AgFTA”by “AgOTf”. Similar error was found in line 260.

This was done (see revised version)

  • Scheme 17 and line 237, whether should the ligand of “dtpby”be “dtbpy”?

This is correct

  • Line 264, please using the bold form for compound of “76”. Similar error was found in line 295 for compound of “81”.

This was done (see revised version)

  • Line 326, please replacement of “fluor”by “fluoride”. Similar error was found in line 345.

This was done (see revised version)

  • Line 348, please revising “ml”by “mL”.

This was done (see revised version)

  • Line 368, please replacement of “idoaryl”by “iodoaryl”.

This was done (see revised version)

  • Line 391, please replacement of “an electron withdrawing groups”by “an electron-withdrawing group”.

This was done (see revised version)

  • Line 431 and Scheme 35, “AgTFA”should be “AgOTf”.

This is correct

  • Line 436, please removal of the blank between “direct”and “arylation”.

This was done (see revised version)

  • Line 513, removal of the blank between “using”and “Ph–Br”, and replacement of “Pd2dba3”by “Pd2(dba)3”.

This was done (see revised version)

  • Line 566, please revising the “Cu(OAc)2.3H2O”. Similar error was found in Line 593, 612 and 618, and Schemes 49 and 50.

This was done (see revised version)

(26) Lines 681–691, please showing the compounds 258, 259, 261–263 by the bold form.

This was done (see revised version)

(27) Page 33, Table 1, please revising “Yield%”by “Yield (%)”.

This was done (see revised version)

(28) Line 787???, it seems that some terms are missing. Please revising this sentence. 

This section was deleted following the request of the reviewer 1.

Round 2

Reviewer 1 Report

Although authors tried to address the specific comments made during first revision, it doesn't look like authors really paid attention to the spelling mistakes. I have observed many spelling and grammatical errors throughout the manuscript, the authors need to carefully go through the manuscript to check for the errors.

In my original comments to authors, I asked to specify or address any electronic or steric effects on the reactivity. The authors response stating "In this case, no notable influence of the substrate substituents on the reactivity" numerous times in the manuscript does not make sense and needs to be addressed properly. 

Even with the addressed comments, in my opinion the manuscript needs a rewrite or language editing before the publication

Author Response

Although authors tried to address the specific comments made during first revision, it doesn't look like authors really paid attention to the spelling mistakes. I have observed many spelling and grammatical errors throughout the manuscript, the authors need to carefully go through the manuscript to check for the errors.

Thank you again dear reviewer for the time you spent reviewing our paper, the manuscript was checked carefully with an English native speaker. The revisions made to the manuscript was marked up using the “Track Changes” function (See revised version).

In my original comments to authors, I asked to specify or address any electronic or steric effects on the reactivity. The authors response stating "In this case, no notable influence of the substrate substituents on the reactivity" numerous times in the manuscript does not make sense and needs to be addressed properly. 

We checked all the manuscript and we found that the sentence “In this case, no notable influence of the substrate substituents on the reactivity" only one time. This sentence was addressed properly and more information was added (See revised version).

Even with the addressed comments, in my opinion the manuscript needs a rewrite or language editing before the publication

We hope that the corrections and modifications we added during these last five days will convince you. Thank you again for your time.